# Efficacy, safety, and economic evaluation of Ojeok-san plus Saengmaek-san for gastroesophageal reflux-induced chronic cough: Protocol for a randomized, double-blind, placebo-controlled, parallel, multicenter, investigator-initiated clinical trial

Yee Ran Lyu[1], O-Jin Kwon[1], Ae-Ran Kim[2], Min Ji Kim[2], Yoon Jae Lee[3], Kwan-Il Kim[4], Jae-Woo Park[5], Seok-Jae Ko[5], Yang-Chun Park[6], Jun-Yong Choi[7], Tae-Yong Park[8], Jun-Hwan Lee[1,9,10]*, Beom-Joon Lee[4]*

1 Korean Medicine Science Research Division, Korea Institute of Oriental Medicine, Daejeon, Republic of Korea, 2 Clinical Medicine Division, R&D Strategy Division, Korea Institute of Oriental Medicine, Daejeon, Republic of Korea, 3 Jaseng Spine and Joint Research Institute, Jaseng Medical Foundation, Seoul, Republic of Korea, 4 Division of Allergy, Immune and Respiratory System, Department of Internal Medicine, College of Korean Medicine, Kyung Hee University, Seoul, Republic of Korea, 5 Department of Gastroenterology, College of Korean Medicine, Kyung Hee University, Seoul, Republic of Korea, 6 Division of Respiratory Medicine, Department of Internal Medicine, College of Korean Medicine, Daejeon University, Daejeon, Republic of Korea, 7 Department of Korean Internal Medicine, Korean Medicine Hospital of Pusan National University, Busan, Republic of Korea, 8 Institute for Integrative Medicine, Catholic Kwandong University International St. Mary's Hospital, Incheon, Republic of Korea, 9 Department of Korean Convergence Medical Science, KIOM School, University of Science & Technology (UST), Daejeon, Republic of Korea, 10 Athinoula A. Martinos Center for Bioimedical imaging, Department of Radiology, Massachusettes General Hospital, Harvard Medical School, Boston, Massachusetts, United States of America

* omdjun@kiom.re.kr/ jlee382@mgh.harvard.edu (JHL); franchisjun@naver.com (BJL)

## Abstract

### Introduction

Gastroesophageal reflux-induced chronic cough (GERC) is a common extraesophageal manifestation of gastroesophageal reflux disease (GERD). However, the mechanisms underlying GERC remain unclear, and current treatments with anti-reflux drugs do not provide significant benefits in the management of GERC. Therefore, safe and effective drugs to treat GERC are urgently needed.

### Methods and analysis

We designed a randomized, double-blind, placebo-controlled, parallel, multi-center, investigator-initiated clinical trial to assess the efficacy, safety, and economics of combined Ojeok-san (OJS) and Saengmaek-san (SMS) in treating GERC. Our trial will be conducted in five hospitals in Korea, and a total of 138 participants will be enrolled, equally divided between the OJS plus SMS and placebo groups. All

**Data availability statement:** No datasets were generated or analysed during the current study. All relevant data from this study will be made available upon study completion.

**Funding:** This research was supported by a grant from the Korea Health Technology R&D Project through the Korea Health Industry Development Institute (KHIDI), funded by the Ministry of Health and Welfare, Republic of Korea (grant number: RS-2022-KH127464).

**Competing interests:** The authors declare that this study was conducted in the absence of any commercial or financial relationships that could be construed as potential conflicts of interest.

participants will be instructed to receive OJS plus SMS or a placebo for 6 weeks and visit hospitals every 2 weeks until week 8 to evaluate their efficacy or safety outcomes. For efficacy outcomes, the cough diary score, cough VAS, Leicester Cough Questionnaire, Gastroesophageal Reflux Disease Questionnaire, Hull Airway Reflux Questionnaire, and 5-level EuroQol 5-dimensional Questionnaire will be evaluated to observe symptoms of cough and GERD, as well as the quality of life in patients with GERC. Pattern identification for the Chronic Cough Questionnaire and gastroesophageal reflux disease will be measured as an additional exploratory outcome. Safety will be assessed in terms of laboratory tests, vital signs, and adverse events; economic evaluation will be simultaneously conducted through the healthcare system and societal perspectives by estimating cost-utility and cost-effectiveness.

## Conclusion

Our study proposes a combination of OJS and SMS to manage the symptoms of GERC as a new insight and this study results will provide scientific evidence for the use of OJS plus SMS in the treatment of GERC.

## Introduction

Gastroesophageal reflux-induced chronic cough (GERC) is among the most common extraesophageal manifestations of gastroesophageal reflux disease (GERD), with cough being observed in 30.5–34.9% of patients with GERD [1]. GERD is also a major cause of chronic cough, along with the upper airway cough syndrome and cough-variant asthma [2]. GERD and coughing affect each other, aggravating GERC. Under the current understandings, the mechanism by which gastroesophageal reflux induces cough is explained by reflux theory (micro-aspiration of the gastric refluxate into the respiratory tract), reflex theory (the stimulation of the esophageal-bronchial cough reflex mediated by the afferent nerves in the distal esophagus) [3], and esophageal dysmotility [4]. However, a causal relationship between GERD and coughing has not yet been established.

Anti-reflux treatments of GERC, such as proton pump inhibitors (PPIs) and H2 antagonists, alone or in combination with promotility agents, are currently considered the standard therapy for GERC [3]. However, recent studies have reported that more than one-third of patients do not respond to PPIs [5], and multiple reviews have found a lack of evidence regarding the use of empirical PPI therapy for GERC [6]. These refractory cases of GERC are often observed in patients who do not have GERD symptoms such as heartburn and regurgitation, and are presumed to be due to non-acidic or weakly acidic reflux (4), which accounts for 80% of PPI-treated chronic cough cases [7]. Recently, for refractory GERC, neuromodulators such as gabapentin and baclofen have been used in some studies; however, their efficacy remains unclear with regard to unnecessary adverse effects [8]. Therefore, further studies are needed to develop a new, effective, and safe drug to manage GERC, including non-symptomatic refractory GERC.

In this study, we propose the herbal medicines Ojeok-san (OJS) plus Saengmaek-san (SMS) for the treatment of GERC as a new insight. Contrary to anti-reflux drugs, which have limitations in non-symptomatic GERC, we targeted the management of the cough reflex by modulating neurogenic airway inflammation and improving esophageal dysmotility. OJS and SMS are widely used in Korean Medicine to manage digestive, respiratory, and neurological disorders owing to their pharmacological activities [9,10]. OJS exerts anti-inflammatory effects by decreasing the levels of pro-inflammatory cytokines [11] and analgesic effects by ameliorating visceral and somatic nociception [12]. It has also been reported to reduce airway inflammation and pulmonary fibrosis, by suppressing the level of T helper type 2 cytokines, VEGF, and TGF-β1/Smad3 expressions [13]. SMS, primarily used for dry cough, is known for its mucolytic effects on the respiratory tract [14], antioxidant and anti-inflammatory effects [10], and gastroprokinetic effects by regulating gastrointestinal (GI) motility and increasing the activity of the interstitial cells of Cajal [15]. Taken together, we expect that OJS plus SMS can attenuate airway inflammation and regulate esophageal motility, thereby exerting its effects on GERC.

We found the potential effects of OJS plus SMS for GERC in our pilot study by showing significant differences in cough diary scores (CDS) compared to the placebo [16]. By addressing the limitations of a previous pilot study, we designed a confirmatory, large-scale, multi-center clinical trial to evaluate the efficacy, safety, and economic value of OJS plus SMS in the treatment of GERC patients. We specified the diagnosis of GERD and the restricted conditions that could affect its safety and efficacy in patients with GERC. Secondary efficacy outcomes evaluating GERD-related symptoms, quality of life, and economic status were added to this trial. Overall, cough symptoms, airway hypersensitivity, GERD-related symptoms, and quality of life will be evaluated in this trial, and our study results will provide evidence for the use of OJS plus SMS in the treatment of GERC.

## Method and analysis

### Trial design

This study is a randomized, double-blind, placebo-controlled, parallel, multi-center, investigator-initiated clinical trial to assess the efficacy, safety, and economic value of OJS plus SMS compared to placebo in patients with GERC. The trial will be conducted at five hospitals in South Korea: Kyung Hee University Korean Medicine Hospital, Gangdong Kyung Hee University Korean Medicine Hospital, Daejeon University Daejeon Korean Medicine Hospital, Pusan National University Korean Medicine Hospital, and Catholic Kwangdong University International St. Mary's Hospital. After participants voluntarily signed a written informed consent, they will be screened for eligibility assessment, and those who meet the inclusion criteria will be enrolled for the next 8 weeks of the trial period (Fig 1). A total of 138 participants will be enrolled in this study. Recruitment of participants is started from January, 2024, and anticipated to be completed by December 2026, and data collection and result analysis are also expected to be completed by the first half of 2027. Participants will be allocated to either the OJS plus SMS group or the placebo group in a 1:1 ratio and will be asked to administer the investigational medicine three times a day for 6 weeks. For the evaluation of efficacy, safety, and economic assessments, participants will visit every 2 weeks until the last follow-up visit on week 8. Detailed trial procedures are presented in Fig 2.

The study protocol will be conducted in accordance with the Declaration of Helsinki and Good Clinical Practice guidelines [17] and complied with the Standard Protocol Items: Recommendations for Interventional Trials (SPIRIT) and Consolidated Standards of Reporting Trials guidelines (CONSORT).

### Participants

**Inclusion criteria.** Participants who meet the following criteria will be enrolled in this trial:

1) Aged 19–65 years

2) Subjects who have had a history of cough continuously for > 8 weeks

| | STUDY PERIOD | | | | | |
|---|---|---|---|---|---|---|
| | Enrolment | Allocation | Post-allocation | | | Follow-up |
| **VISIT** | *1* | *2* | *3* | *4* | *5* | *6* |
| **TIMEPOINT** | *-2~1 week* | *0* | *2 weeks ± 3days* | *4 weeks ± 3days* | *6 weeks ± 3days* | *8 weeks ± 3days* |
| **ENROLMENT:** | | | | | | |
| Eligibility screen | X | | | | | |
| Informed consent | X | | | | | |
| Demographics | X | | | | | |
| Medical and treatment history | X | | | | | |
| Physical examination | X | | | | | |
| Vital sign | X | X | X | X | X | X |
| Chest X-ray | X | X | X | X | X | X |
| Pulmonary Function Test | X | | | | | |
| FeNO | X | | | | | |
| Nasal Endoscopy | X | | | | | |
| PNS x-ray (if needed) | X | | | | | |
| EKG | X | | | | | |
| Laboratory tests | X | (X) | | | X | |
| Pregnancy Test | X | | | | | |
| Allocation | | X | | | | |
| **INTERVENTIONS:** | | | | | | |
| OJS plus SMS | | ←———————————→ | | | | |
| Placebo | | ←———————————→ | | | | |
| **ASSESSMENTS:** | | | | | | |
| Cough Symptom Score | X | X | X | X | X | X |
| Cough VAS | X | X | X | X | X | X |
| LCQ-K | | X | X | X | X | X |
| GERDQ | | X | X | X | X | X |
| HARQ | | X | X | X | X | X |
| Pattern Identification for Chronic Cough Questionnaire | X | X | | | X | |
| Pattern Identification for GERD | | X | | | X | |
| EQ-5D-5L | | X | X | X | X | X |
| Abdominal Examination | | X | | | | |
| WPAI:SHP | | X | X | X | X | X |
| Adverse events | | X | X | X | X | X |
| Compliance test | | | X | X | X | |

FeNO; fractional exhaled nitric oxide, EKG; Elektrokardiogramm, PNS; paranasal sinuses, VAS; visual analog scale, LCQ-K; Leicester Cough Questionnaire – Korean Version, GERDQ; Gastroesophageal reflux disease questionnaire, HARQ; Hull Airway Reflux (hypersensitivity) Questionnaire, EQ-5D-5L; 5-level EuroQol 5-dimensional questionnaire, WPAI:SHP; Work Productivity and Activity Impairment Questionnaire: Specific Health Problem

**Fig 1. Flowchart of the trial process.**

3) Subjects who had been diagnosed with GERD within the last 1 year (those who submitted documents diagnosed with reflux esophagitis at other hospitals or prescribed reflux esophagitis medicines for more than 4 weeks)

4) Subjects who consent to participate

**Exclusion criteria.** Participants who meet at least one of the following criteria will be excluded:

1) Present with abnormal findings on chest radiography, pulmonary function test, or nasal endoscopy that might lead to cough.

2) Diagnosis of acute respiratory diseases (including upper respiratory tract disorders) within the previous month.

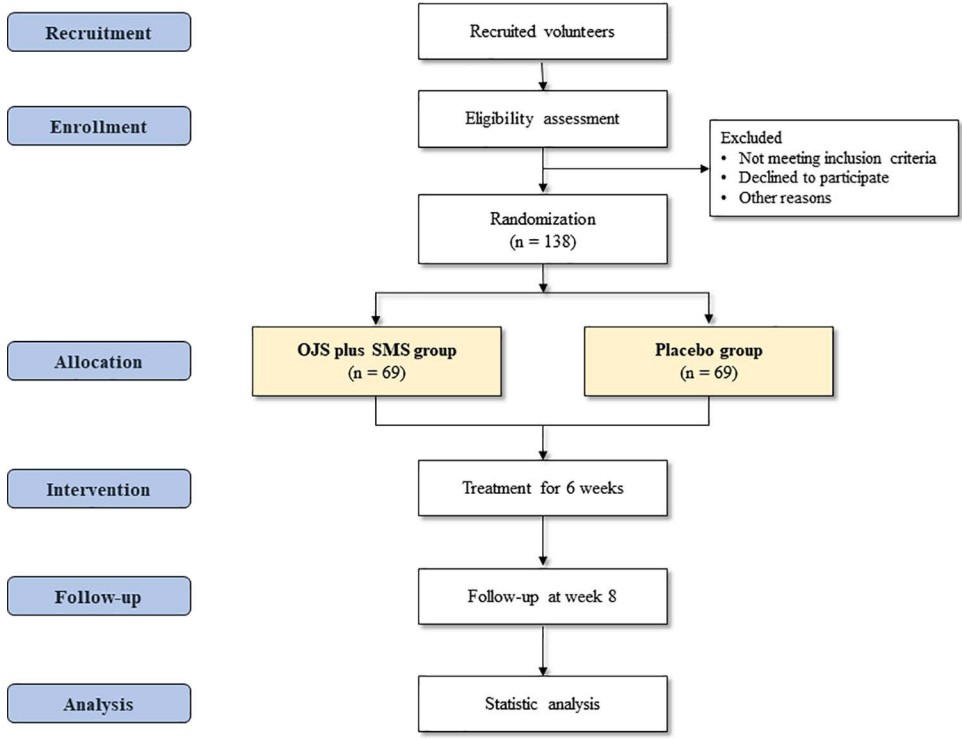

**Fig 2. Schedule of enrolment, interventions, and assessments.**

3) Presence of postnasal drip syndrome

4) Diagnosis of chronic respiratory diseases (e.g., chronic obstructive pulmonary disease, bronchial asthma, bronchiectasis, interstitial lung disease, and other chronic respiratory diseases) within the last 2 years

5) Diagnosed with Los Angeles classification system grade C or higher GERD within the past year

6) Symptoms indicative of malignant disease within the GI tract (e.g., severe dysphagia, bleeding, weight loss, anemia, and bloody stools)

7) History of esophagostenosis, esophageal varices, Barrett's esophagus, peptic ulcer or gastrointestinal bleeding, and Zollinger-Ellison syndrome.

8) History of surgical or endoscopic anti-reflux treatment

9) Treatment with angiotensin-converting enzyme inhibitor during the previous 4 months

10) Treatment with cough medicines, glucocorticoids, leukotriene receptor antagonists, anticholinergic drugs, long-acting β2-agonists within the previous 2 weeks

11) Treatment with antihistamines within the previous 2 weeks

12) Treatment with PPIs, histamine receptor antagonists, mucosa-protective agents, GI motility promoters, antacids, antidepressants, anxiolytics, and lower esophageal sphincter agonists within the previous 2 weeks

13) Treatment with digitalis, mineralocorticoids, anticoagulants, and high-dose aspirin (≤325mg is allowed) within the previous 4 weeks

14) Treatment with potassium-containing drugs, licorice-containing drugs, glycyrrhizinic acid or its salt-containing drugs, loop diuretics (furosemide, ethacrynic acid), or thiazide diuretics (trichloromethiazide) within the previous four weeks

15) A lifetime smoking history of ≥20 packs (400 cigarettes)

16) Body mass index < 18.5 kg/m$^2$

17) Aspartate aminotransferase (AST) or alanine aminotransferase (ALT) level at least twofold higher than the upper limit of normal or an eGFR ≤ 60 mL/min/1.73m$^2$

18) Unregulated hypertension (high blood pressure of 160 mmHg in the condenser or high blood pressure exceeding 100 mmHg in a relaxation period > 3 min)

19) Active infection requiring systemic antibiotic therapy

20) Blood-clotting disorder

21) Hepatitis B (active) or hepatitis C, chronic liver disease

22) History of cerebrovascular accident (CVA)

23) History of malignant tumors (e.g., lung or esophageal cancer) within the last five years

24) History of drug or alcohol abuse

25) Allergies or sensitivities to the experimental medicine/placebo

26) Pregnant or breastfeeding

27) Subjects who did not agree to use contraception by medically permitted methods

28) Subjects who have participated in clinical trials within the past month

29) Subjects who are judged by the investigators to be inappropriate for the clinical trial

**Sample size.** Based on our previous study results, we calculated a sample size of 138 (69 participants for each group) for this trial [16]. The calculation was conducted to test the difference in the change in CDS from baseline (week 0) to post-dosing of OJS plus SMS (week 6) compared with the placebo. In our preliminary clinical trial, the mean difference between the two groups was found to be 0.75 and the standard deviation was 1.4 [16]. We assumed the power to detect the difference to be 0.8, the two-sided significance level to be 0.05, and a dropout rate of 20% to determine the sample size.

**Recruitment.** Participants will be recruited from the outpatient departments of Pulmonology or Gastroenterology at each affiliated hospital. Recruitment brochures will be posted on the bulletin boards and websites of each hospital, and advertisements in mass media or local communications will be conducted as needed to promote recruitment.

## Randomization and blinding

An independent statistician conducted randomization using a computer random number generator of SAS® Version 9.4 (SAS institute. Inc., Cary, NC, USA) using the block randomization method. The allocation ratio was 1:1 for each intervention (OJS plus SMS) and control (placebo) group. The generated random sequence was transmitted to the manufacturer responsible for labeling the participants' identification codes on each investigational medicine package. These labelled medicines will be delivered to management pharmacists at each hospital, and pharmacists will give the corresponding investigational medicine to each participant using their identification code. The random number will be kept by an

independent investigator who has no relationship with the patient throughout the trial and will only be disclosed when a serious adverse event occurs.

All investigators, participants, and outcome assessors will be blinded. The intervention and placebo drugs are distinguished only by the participants' identification codes, labelled in identical and opaque packages, and both drugs were manufactured with the same color, taste, and smell.

### Intervention

**OJS plus SMS.** OJS, comprising 17 herbs, is composed of 4.35 g of granules per package, and SMS, consisting of three herbs, is composed of 1.31 g of granules per package. The participants will be instructed to administer OJS plus SMS (5.86g/each) three times a day for 6 weeks. The detailed composition of each drug is presented in Table 1. Both OJS and SMS are insurance-covered Korean medicine granules, and the dosage of each drug was based on the approved dosage of the Ministry of Food and Drug Safety (MFDS) of Korea. OJS and SMS were manufactured by Han Kook Shin Yak Pharm Co. Ltd. (Nonsan, Chungnam, South Korea) and packaged by the National Institute of Korean Medicine Development (Gyeongsan, Gyeongbuk, South Korea), both of which obtained authorization from Korea Good Manufacturing.

**Placebo.** The placebos for OJS and SMS were manufactured and packaged by the National Institute of Korean Medicine Development in accordance with the guidelines on placebos. Placebos do not contain any of the active ingredients, OJS or SMS, and are composed of starch, lactose, and coloring and flavoring agents. Both placebos were developed to have a similar color, taste, and smell to the intervention drugs.

**Table 1. Components of Ojeok-san and Saengmaek-san.**

| Herb name | Latin name | Amount (g) |
|---|---|---|
| *Ojeok-san (OJS)* | | |
| Changchul | Atractylodis Rhizoma | 0.95 |
| Mahwang | Ephedrae Herba | 0.2 |
| Jinpi | Citri Unshius Pericarpium | 0.4 |
| Hubak | Magnoliae Cortex | 0.08 |
| Gilgyeong | Platycodonis Radix | 0.43 |
| Jigak | Aurantii Immaturus Fructus | 0.31 |
| Danggwi | Angelicae Gigantis Radix | 0.37 |
| Geongang | Zingiberis Rhizoma | 0.22 |
| Jagyak | Paeoniae Radix | 0.27 |
| Bongnyeong | Poria Sclerotium | 0.02 |
| Cheongung | Cnidii Rhizoma | 0.3 |
| Baekji | Angelicae Dahuricae Radix | 0.31 |
| Banha | Pinelliae Tuber | 0.22 |
| Yukgye | Cinnamomi Cortex | 0.04 |
| Gamcho | Glycyrrhizae Radix et Rhizoma | 0.2 |
| Saenggang | Zingiberis Rhizoma Recens | 0.03 |
| Total | | 4.35 |
| *Saengmaek-san (SMS)* | | |
| Maengmundong | Liriopis seu Ophiopogonis Tuber | 0.75 |
| Insam | Ginseng Radix | 0.30 |
| Omija | Schisandrae Fructus | 0.36 |
| Total | | 1.41 |

## Outcomes

**Primary efficacy outcome. Cough Diary Score (CDS):** The CDS is a subjective cough score questionnaire that assesses the severity and frequency of cough on five scales (0–4) during the daytime and nighttime [18]. The cough severity scale is divided into 0: none, 1: slight, 2: mild, 3: moderate, and 4: severe, whereas the cough frequency ranges between 0 (none), 1 (infrequent/occasional), 2 (several times), 3 (many times), and 4 (all the time). Each of the daytime and nighttime scores is calculated by adding the score of cough severity and cough frequency, with a maximum score of 8; the total CDS is an averaged score of daytime and nighttime CDS. All participants will be instructed to record their CDS twice a day: daytime score assessing symptoms occurring from 8am to 8 pm and nighttime score from 8 pm to 8am. At every visit, a Cough Diary will be given to each participant to record their symptoms until the next visit, and the average score between visits will be used as the outcome to evaluate the efficacy of OJS plus SMS compared with the placebo.

The primary outcome of this trial is the change in CDS between baseline (pre-drug) and week 6 (post-drug) in the OJS plus SMS group compared to the placebo group. CDS was chosen as the primary outcome measure, as a subjective cough score is the most frequently used outcome in clinical trials related to cough [19] and was also the primary outcome in our pilot trial showing significant differences between OJS plus SMS and placebo.

**Secondary efficacy outcome. Cough VAS:** The cough VAS is another cough score outcome widely used to assess acute or chronic cough [20,21]. It is also highly responsible for changes in cough severity, making it easy to compare therapeutic effects in clinical trials [22]. The participants were asked to mark the severity of cough on the line with a calibration of 0–100 mm, with 0 indicating 'no cough' and 10 indicating 'unbearable cough'. The distance between zero and the marked point will be measured using the cough visual analog scale score. Participants will be instructed to record the cough VAS daily in the given cough diary for the next 2 weeks, and the average score will be used as the outcome.

**Leicester Cough Questionnaire – Korean version (LCQ-K):** The LCQ-K, which consists of 19 items, is a quality of life (QoL) questionnaire specified for cough symptoms. Each item ranges from 1 to 7, with higher scores indicating a better quality of life and is divided into three domains: physical, mental, and social [23]. Previous studies have demonstrated a relationship between the LCQ score and the severity of cough, and it is also recommended by the European Respiratory Society as a QoL outcome for the impact of cough in patients with chronic cough [24]. The validated Korean version of the LCQ will be used in the trials at each visit [25].

**Gastroesophageal reflux disease questionnaire (GERDQ):** The GERDQ is a validated patient-report questionnaire for diagnosis and management of GERD [26]. It includes six items related to symptoms such as heartburn, regurgitation, upper stomach pain, nausea, night sleep disturbance, and the need for additional medication. Each item is graded from 0 to 3 according to the frequency of symptoms in the past 1 week, with a higher score indicating more severe GERD-related symptoms. As the GERDQ is not only used for assessing changes in treatment outcomes but also for diagnosing GERD, it will be useful to identify GERD severity in patients enrolled in our trial as none, mild, moderate, or severe [27]. Therefore, it can be used to compare the efficacy of OJS plus SMS in symptomatic and non-symptomatic patients with GERD.

**Hull Airway Reflux Questionnaire (HARQ):** The HARQ is designed to measure cough hypersensitivity and contains 14 items graded from 0 to 5, with a maximum score of 70 points. Higher scores indicated higher sensitivity of the airway [28]. As cough hypersensitivity is highly related to chronic cough and represents a distinct clinical entity, the HARQ was used to assess patients with GERC in this trial.

**5-level EuroQol 5-dimensional questionnaire (EQ-5D-5L):** The EQ-5D-5L is a self-report questionnaire for measuring generic health status established by the EuroQol Group and is one of the most used QoL questionnaires in a wide range of populations [29]. The domains are divided into five parts, including mobility, self-care, usual activity, pain/discomfort, and anxiety/depression, and each part is scored on five levels. Additionally, participants rated the scale lines marked from 0 mm (the worst health you can imagine) to 100 mm (the best health you can imagine). The EQ-5D-5L was also used for economic evaluation.

**Exploratory outcomes. Pattern Identification for Chronic Cough Questionnaire (PICCQ):** The Pattern Identification for Chronic Cough Questionnaire will be assessed at baseline to compare the differences in efficacy by pattern identification. Based on the pattern identification for chronic cough, participants will be classified into one of five patterns: cold wind, phlegm turbidity, fire heat, lung deficiency, or kidney yang deficiency [30]. These pattern identifications are the diagnostic and treatment criteria used in Korean Medicine to determine appropriate therapies for each patient. Thus, it is important to assess the relationship between these patterns and the therapeutic efficacy of OJS plus SMS. **Pattern Identification for GERD:** Pattern identification for GERD will also be performed in all participants at baseline. We will investigate the distribution of pattern identification in patients with GERC and the pattern identification that best responds to OJS plus SMS. Pattern identification of GERD includes stagnation of Liver Qi, stomach yin deficiency, spleen–stomach weakness, and spleen-stomach dampness-heat. The questionnaire contains 40 items, of which 23 are self-report items and 8 are assessed by a Korean Medicine Doctor [31].

**Safety outcomes.** For the safety outcomes, vital signs (blood pressure, body temperature, pulse rate, and respiratory rate), adverse events, and laboratory test will be evaluated during the trial period. At every visit, vital signs and adverse events will be assessed, and laboratory tests will be conducted before and after drug administration. Liver function tests (aspartate aminotransferase, alanine aminotransferase, alkaline phosphatase, gamma-glutamyl transferase, bilirubin, and protein/albumin), renal function tests (blood urea nitrogen and creatinine), electrolyte content (sodium, potassium, and chloride), blood coagulation tests (prothrombin time and activated partial thromboplastin time), and complete blood count will be performed. Adverse events will be collected by asking participants whether any unfavorable and unintended signs, symptoms, or diseases presented after taking the investigational medicine. These will then be assessed for severity and causality of investigational drugs, and investigators will follow up until the adverse events end.

**Economic evaluation.** We will conduct an economic evaluation from the healthcare system and societal perspective over 8 weeks of the trial period in parallel with a randomized controlled trial (RCT). We will estimate cost-utility as the primary economic endpoint and cost-effectiveness as a secondary economic endpoint of OJS plus SMS compared with placebo. Utility data will be collected from trial outcomes of the EQ-5D-5L, a health-related quality of life questionnaire assessed at every visit, and quality-adjusted life years (QALYs) will be calculated to estimate the incremental cost-utility ratio. Effectiveness data will also be obtained from the efficacy outcomes in the main RCT, including the CDS, cough VAS, LCQ-K, GERDQ, and HARQ scores, which will be used to estimate the incremental cost-effectiveness ratio. Cost data will be obtained in terms of medical and non-medical costs (transportation, time, and nursing care costs) and productivity loss costs from institutional data, a specially developed questionnaire for cost, and data from Statistics Korea, if needed. Additionally, sensitivity analysis will be performed on possible variables using one-way sensitivity analysis and probability sensitivity analysis with representative values and the distribution of variables.

## Statistical analysis

Statistical analysis will be performed by an independent statistician using SAS Analytics Pro software and significance will be accepted at an α-level of 0.05 in a two-sided test. For all data analyses, full analysis (FAS) set will be the primary analysis set in this trial, and a per-protocol (PP) set will be performed as an additional analysis. In this trial, the FAS set included all randomly assigned participants who had been evaluated for efficacy at least once after the administration of the investigational medicine. PP sets were defined as participants who completed the trial without violating the protocol, with compliance lower than 75%, taking prohibited medications, or presenting with serious adverse events. Continuous variables are presented as means and 95% confidence intervals and categorical variables as frequencies and percentages.

For primary efficacy analysis, the changes in CDS between baseline and week 6 of OJS plus SMS compared with placebo will be evaluated using a mixed-effect model repeated measure (MMRM), with each group and visit set as fixed effects and participants as random effects. Secondary efficacy analysis, comparing cough VAS, LCQ-K, GERDQ, HARQ,

and EQ-5D-5L evaluated at each visit point with baseline between groups, will also be performed using MMRM. The analysis of the changes in each efficacy outcome within the group will be performed using a paired t-test or Wilcoxon signed-rank test, depending on its normality. For missing values, a multiple imputation will be performed. Additionally, to analyze the trends of CDS over time compared with the baseline, RM-ANOVA will be conducted, and Dunnett's procedure (based on the baseline) will be used as a multiple comparison correction. A safety analysis will be conducted to compare the number of adverse events between groups using Fisher's exact test.

## Data collection, management, and monitoring

The principal investigator should manage all clinical trial data in compliance with Good Clinical Practice guidelines and related regulations regarding collection, recording, and reporting. Clinical data collected throughout the trial will be entered into an electronic Case Report Form (eCRF) and managed by a data manager. All source documents must be stored in a locked place and security must be maintained by storing them on a computer with limited access to those not related to the trial. These documents will be preserved for 3 years from the end of the clinical trial.

When recording a participant's personal information, all data will be handled in accordance with the relevant regulations to ensure confidentiality. All records identifying the personal information of all participants will be managed and evaluated according to the participants' identification numbers and the initials assigned at the start of the study. Only the monitor, the person conducting the inspection, the review committee, or the Minister of Food and Drug Safety can view medical records and clinical trial data within the scope set by relevant regulations without violating test subject confidentiality.

Data monitoring will be conducted during regular and occasional visits by a clinical research associate (CRA) at the Korea Institute of Oriental Medicine. Compliance with the clinical trial protocol, collection of appropriate and accurate data, inspection of informed consent forms, collection and reporting of (serious) adverse reactions, and management of investigational drugs will be confirmed and inspected according to the monitoring plan.

## Ethics approval and dissemination

The study protocol(Ver. 1.2) was authorized by the Ministry of Food and Drug Safety of Korea (MFDS) (approval number 101726) and approved by the Institutional Review Board of Kyung Hee University Korean Medicine Hospital (KOMCIRB2022-11-006), Gangdong Kyung Hee University Korean Medicine Hospital (KHNMCOH2023-10-004), Daejeon University Daejeon Korean Medicine Hospital (DJDSKH-23-DR-10), Pusan National University Korean Medicine Hospital (PNUKHIRB2023-10-003), and Catholic Kwandong University International St. Mary's Hospital (IS23EIME0065). This study was registered with the Clinical Research Information Service (KCT0008908).

Prior to trial screening, all participants will be provided with all information related to the clinical trial with sufficient time and opportunity to ask questions, decide whether to participate, and will be asked to sign written informed consent. They will also be promptly informed of any changes in the clinical trial-related information, even during the trial period. After the trial ends, we will disseminate the study results through peer-reviewed journals, conference presentations, and clinical research information services.

## Discussion

Along with the growing prevalence of GERD worldwide, the management of chronic cough due to GERD has emerged as an important clinical issue. Similar to GERD, anti-reflux drugs, along with promotional agents, are used to treat GERC to suppress the acidity of refluxates or promote esophageal clearance [3]. However, anti-reflux agents have provided only moderate benefits in patients with GERD symptoms and no significant benefits in non-symptomatic patients, accounting for up to 75% of patients with GERC [3]. Until now, neuromodulators proposed to regulate transient lower esophageal sphincter relaxation (TLESRs) also have limited evidence in GERC use [5]. These refractory responses are thought to be

due to the complicated mechanisms underlying GERC, which are associated with one or more of the following physiologies: non-acid reflux, vagal reflex, airway inflammation, and esophageal or cough hypersensitivity [32]. To manage these complex etiologies of GERC, the use of more than one agent is another strategy to achieve preferable effectiveness.

From this perspective, we propose a combination of OJS and SMS to manage the symptoms of GERC, expecting OJS to regulate GI motility and SMS to attenuate airway inflammation, thereby OJS plus SMS can improve the symptoms of cough induced by GERD. In our previous trial, OJS plus SMS showed promising therapeutic effects in patients with GERC by significantly relieving cough compared to placebo [16]. Based on these preliminary study results, we designed a large-scale, multi-center trial to confirm the efficacy and safety of OJS plus SMS compared to a placebo. These preliminary data are one of our study's strengths in that we could design our trial more accurately, with a higher probability of detecting the efficacy of OJS plus SMS. Our sample size was calculated using the exact effect size, which was revealed in a pilot study, and we supplemented the outcome measures to evaluate the efficacy and safety of OJS plus SMS. Additionally, we added four more hospitals for the successful recruitment of participants to resolve the difficulty in recruitment in our previous study and restricted some of the conditions or medications that could affect the efficacy and safety of outcomes by excluding diseases such as chronic liver disease, cerebrovascular accident, blood-clotting disorder, and unregulated hypertension, and medicines such as digitalis, mineralocorticoid, anticoagulants, and high-dose aspirin. Another strength of our trial design is that we targeted patients with GERC with and without esophageal reflux symptoms, which may be helpful in understanding the physiology of GERC by comparing whether the efficacy of OJS plus SMS is different in symptomatic and non-symptomatic patients. Moreover, our study results will be valuable if significant benefits are proven in non-symptomatic patients with GERC, who are usually revealed to have refractory responses to anti-reflux drugs and have no recommended therapies to manage cough. These results may also be helpful in understanding the physiology of GERC. Lastly, we will perform abdominal examinations and pattern identification for chronic cough and GERD to investigate differences in efficacy depending on the abdominal conditions or pattern identification. These are some of the major diagnostic and management criteria used in Korean Medicine to choose appropriate medications and evaluate the symptoms or diseases in each patient [33]. Our pilot study also found that OJS plus SMS showed the greatest improvement in the GERD pattern types of Liver Qi stagnation. By sub-analyzing the outcomes based on these diagnostic tools, we expect to obtain additional clinical data on abdominal examinations and pattern identification.

Our study is also meaningful in that it is the first clinical trial to investigate the efficacy and safety of herbal medicines for reflux-related chronic cough and our pilot study results has been reported as one of several studies providing new insights on treatment of GERC in a recent review paper [32]. Our results provide scientific evidence supporting the use of herbal medicines in patients with GERC. Moreover, this is the first clinical trial to use a combination of herbal medicines, both of which are included in 56 types of insurance-covered Korean medicine (KM) granules in Korea. We will also conduct an economic evaluation along with this trial to provide supportive evidence from a healthcare system and societal perspective. This will have significance in the health insurance system of Korean Medicine, as our study design can be used as a basis for prescribing more than two insurance-covered KM granules simultaneously in clinics, which is limited to the Korean health insurance system.

Our study has some limitations. First, 24-hour esophageal pH monitoring or multichannel intraluminal impedance combined with pH monitoring was not used when diagnosing or evaluating the efficacy of GERC. However, these procedures are invasive and have limitations in terms of providing causal evidence [34]. Instead, we followed the American College of Chest Physicians guidelines to diagnose GERC by excluding other potential causes of chronic cough to predict reflux-induced cough, and we added gastrointestinal endoscopy to our inclusion criteria for diagnosis of GERD. Additionally, recent study indicated that GERDQ could be an alternative to multichannel intraluminal impedance-pH monitoring for diagnosis and management of GERC. As GERDQ is used as a secondary efficacy outcome for our study, it will also contribute on diagnosis for GERC. Moreover, our study does not aim to find a relationship between acid or non-acid reflux and the occurrence of cough, but rather to find differences between symptomatic and non-symptomatic patients with

GERC; therefore, the need for these procedures is not essential. Second, we did not consist objective outcomes in our trial, as cough frequency monitoring. Despite the impact of objective cough frequency as a key clinical end-point, existing wearable monitors still have limitations in feasibility and utility in clinical practice settings. We will consider using a validated and continuous cough monitoring in further studies after we demonstrate the efficacy of OJS plus SMS in this trial. Third, the formulation of our investigational medicines is a powder, not a capsule or tablet. This formulation has previously been reported to have a placebo effect in numerous studies [35]. However, as insurance-covered KM granules are mostly manufactured using powder formulas, it is difficult to change their formation, which should be approved by the MFDS in Korea. We will interpret our results by considering these limitations, and further studies will be conducted to address them. Despite these limitations, our study will provide evidence on the efficacy, safety, and economic data of OJS plus SMS in a well-designed clinical trial approved by MFDS and we expect OJS plus SMS to be presented as a new, safe, and effective agent for GERC.

## Conclusion

Our study proposes a combination of OJS and SMS to manage the symptoms of GERC as a new insight. Based on our preliminary study results, we designed a large-scale, multi-center trial to confirm the efficacy and safety of OJS plus SMS compared to a placebo in GERC. We will evaluate cough symptoms, airway hypersensitivity, GERD-related symptoms, and quality of life in this trial, and our study results will provide scientific evidence for the use of OJS plus SMS in the treatment of GERC for the first time.

## Supporting information

**S1 File. Trial protocol.**
(DOCX)

**S2 File. SPIRIT checklist.**
(DOC)

## Author contributions

**Conceptualization:** Kwan-Il Kim.

**Data curation:** Ae-Ran Kim, Min Ji Kim.

**Formal analysis:** O-Jin Kwon, Yoon Jae Lee.

**Investigation:** Jae-Woo Park, Seok-Jae Ko, Yang-Chun Park, Jun-Yong Choi, Tae-Yong Park.

**Project administration:** Jun-Hwan Lee, Beom-Joon Lee.

**Writing – original draft:** Yee Ran Lyu.

**Writing – review & editing:** Beom-Joon Lee.

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
