## [Decision Letter · Decision Letter 0]

Dear Dr. Lyu,

Thank you for submitting your manuscript to PLOS ONE. After careful consideration, we feel that it has merit but does not fully meet PLOS ONE’s publication criteria as it currently stands. Therefore, we invite you to submit a revised version of the manuscript that addresses the points raised during the review process.

We look forward to receiving your revised manuscript.

Kind regards,

Emmanuel O Adewuyi, BPharm, MPH, PhD

Academic Editor

PLOS ONE

 [This research was supported by a grant from the Korea Health Technology R&D Project through the Korea Health Industry Development Institute (KHIDI), funded by the Ministry of Health and Welfare, Republic of Korea (grant number: RS-2022-KH127464).]. 

4. Please match your authorship list in your manuscript file and in the system.

Reviewers' comments:

Reviewer's Responses to Questions

**Comments to the Author**

1. Does the manuscript provide a valid rationale for the proposed study, with clearly identified and justified research questions?

Reviewer #1: Yes

Reviewer #2: Yes

2. Is the protocol technically sound and planned in a manner that will lead to a meaningful outcome and allow testing the stated hypotheses?

Reviewer #1: Yes

Reviewer #2: Yes

3. Is the methodology feasible and described in sufficient detail to allow the work to be replicable?

Reviewer #1: Yes

Reviewer #2: Yes

4. Have the authors described where all data underlying the findings will be made available when the study is complete?

Reviewer #1: Yes

Reviewer #2: Yes

5. Is the manuscript presented in an intelligible fashion and written in standard English?

Reviewer #1: Yes

Reviewer #2: Yes

You may also provide optional suggestions and comments to authors that they might find helpful in planning their study.

Reviewer #1: The study is well-structured and follows a rigorous enough methodological/statistical framework, employing a randomized, double-blind, placebo-controlled design, which is the gold standard for clinical trials. The sample size of 138 participants (69 participants for each group) appears reasonable, as it is based on a power analysis that considers the expected effect size and accounts for potential dropout rates. The statistical methods used to analyze the primary outcome, particularly the mixed-effect model for repeated measures (MMRM), are appropriate for handling longitudinal data, allowing for assessing treatment effects over time while accounting for intra-subject variability. Additionally, the study takes care to minimize bias through a well-executed randomization and blinding procedure, ensuring that neither participants nor investigators are aware of group assignments. The inclusion and exclusion criteria are clearly defined, aiming to reduce confounding factors that could impact the study's validity. The use of multiple imputation for missing data further strengthens the analysis by addressing potential biases introduced by participant dropout or missing values. The selection of primary and secondary outcomes is generally well-justified. While the Cough Diary Score (CDS) is subjective, it is commonly used in clinical trials related to cough assessment. Moreover, including validated secondary outcome measures, such as the Leicester Cough Questionnaire (LCQ-K) and the Gastroesophageal Reflux Disease Questionnaire (GERDQ), adds robustness to the overall evaluation of treatment efficacy. The study also integrates an economic evaluation, which enhances its potential relevance from a healthcare policy perspective.

Major comments

Despite these strengths, the following four methodological aspects, with particular emphasis on the third one, require further attention before the paper can be accepted. Addressing these issues would further strengthen the study’s credibility and ensure that its findings are interpreted with appropriate statistical rigor.

1. The reliance on the Cough Diary Score (CDS), a patient-reported measure, raises concerns about potential bias, even with blinding. It would be beneficial to include more objective measures of cough severity, such as cough frequency monitoring via acoustic analysis.

2. The study does not use 24-hour esophageal pH monitoring or multichannel intraluminal impedance, which are considered more objective measures for diagnosing GERD and its role in chronic cough. This could lead to misclassification bias, as some participants may not have true reflux-induced chronic cough. Justifying this methodological choice and discussing its implications in greater detail would strengthen the study.

3. While the study evaluates multiple secondary outcomes, it is not explicitly stated whether statistical corrections for multiple comparisons (e.g., Bonferroni or false discovery rate adjustments) are applied. Given the number of secondary analyses, failing to account for multiple testing could increase the likelihood of false-positive findings.

4. The sample size calculation is based on an effect size of 0.75, but it is unclear whether this estimate is conservatively justified. If the true effect size is smaller, the study may be underpowered to detect meaningful differences. Providing a sensitivity analysis or justifying the robustness of this assumption would help address this concern.

Reviewer #2: In addressing the prevalent extraesophageal symptom of gastroesophageal reflux disease (GERD), chronic cough (GERC), this manuscript makes a significant contribution to a better application. A total of 138 participants improves the findings' external validity and generalizability across various patient populations and clinical settings. In short, they designed a randomized, double-blind, placebo-controlled, parallel, multi-center, investigator-initiated clinical trial to evaluate the effectiveness, safety, and economics of the combined herbal medicine of Ojeok-san (OJS) and Saengmaek-san (SMS) in treating GERC in five hospitals in Korea. Then, using a number of validated tools, including the cough diary score, cough VAS, and an excessive number of questions, they demonstrated the effectiveness. Furthermore, the study was approved by the IRBs and MFDS of several respectable institutions, guaranteeing ethical rigor. Additionally, the registration (KCT0008908) enhances dependability and transparency. However, there are a few limitations to this study that might be considered to enhance the manuscript for publication:

1- NO LINE NUMBERS/ NO conclusion, so please add a paragraph at the end of discussion to summarize what are the outcomes from this study for clarity.

2- The study makes use of traditional herbal formulations (Ojeok-san and Saengmaek-san), and repeatability may be impacted by the variations in herbal composition.

3- Despite seeking to evaluate safety and efficacy, the study doesn't seem to look at the mechanisms underlying GERC or the methods that OJS+SMS work.

4- Given that GERC can be chronic and variable, a 4- to 6-week course of treatment and an 8-week total follow-up may not be enough to evaluate long-term efficacy, safety, and recurrence of symptoms.

5- Because the study was only carried out in Korea, its applicability to Western populations or other healthcare environments may be limited by cultural, dietary, and medical variations.

6- There is no reference to the use of objective diagnostic tools for GERD/GERC, such as pH monitoring and impedance testing, which could impair the accuracy of determining the severity of the illness and its response to treatment.

**Do you want your identity to be public for this peer review?** For information about this choice, including consent withdrawal, please see our Privacy Policy

Reviewer #1: No

Reviewer #2: No

---

## [Author Response · Author response to Decision Letter 1]

15 Apr 2025

'Response to Reviewers'

Review1:

1. The reliance on the Cough Diary Score (CDS), a patient-reported measure, raises concerns about potential bias, even with blinding. It would be beneficial to include more objective measures of cough severity, such as cough frequency monitoring via acoustic analysis.

-> Thank you for your consideration by suggesting objective measures as cough frequency monitoring, which raise concerns about potential bias for the trial. We had also considered using cough frequency monitoring as an objective measure, as cough frequency monitoring is regarded as an important measure in chronic cough, in which more advance device is being developed and applied on clinical trials. However, existing monitoring devices still have limitations in feasibility and utility for our trial, for that it is not yet used in real-world clinic. Thus, we determined not to use these devices for this trial as our study is the first clinical trial evaluating the efficacy and safety of OJS plus SMS in GERC patients. After we obtain beneficial efficacy by this trial, we will apply these additional objective measures in further studies. These issues regarding outcome measures were also considered during the approval by the Ministry of Food and Drug Safety and IRBs, and were determined considering the real-world application. We once again appreciate your thoughtful considerations on our outcome measures, and hope understand these circumstances of our trial.

2. The study does not use 24-hour esophageal pH monitoring or multichannel intraluminal impedance, which are considered more objective measures for diagnosing GERD and its role in chronic cough. This could lead to misclassification bias, as some participants may not have true reflux-induced chronic cough. Justifying this methodological choice and discussing its implications in greater detail would strengthen the study.

-> Thank you for your recommendation of using multichannel intraluminal impedance pH monitoring, which is known as objective and more accurate diagnostic tool for GERC. As you had mentioned, MII-ph testing are being used to diagnose GERD and its role in chronic cough, thereby can also being used to investigate pharmacological mechanisms. However, we determined MII-ph testing is not essential in our study, as our study does not aim to find a relationship between acid or non-acid reflux and the occurrence of cough, but rather to find differences between symptomatic and non-symptomatic patients with GERC. Moreover, American College of Chest Physicians guidelines (2023) still suggest diagnostic algorithm by excluding other potential causes and we followed this guideline which is also being frequently used in real world clinics. By following guidelines, we had included GERC patients for those who had diagnosed for GERD by endoscopy. We also evaluate GERDQ, which has been reported as an alternative for MII-ph. Although MII-ph testing is more accurate test for diagnosing GERC, following American College of Chest Physicians guidelines or evaluating with GERDQ, can also be used for diagnosis for GERC both on real-world practices or clinical trials without addressing misclassification bias. We had also added the references and explanations for this methodological choice in discussion. Despite these limitations of using MII-ph testing in this trial, we will consider using it on our further trials when evaluating pharmacological mechanisms of OJS plus SMS on GERC. We really appreciate your opinion to strengthen our methodology.

3. While the study evaluates multiple secondary outcomes, it is not explicitly stated whether statistical corrections for multiple comparisons (e.g., Bonferroni or false discovery rate adjustments) are applied. Given the number of secondary analyses, failing to account for multiple testing could increase the likelihood of false-positive findings.

-> Thank you for your consideration on statistical corrections for multiple comparisons. In our study, as stated in ‘2.6 Statistical analysis’, Dunnett’s procedure (based on the baseline) will be used as a multiple comparison correction. As our secondary outcomes are analyzed by changes between baseline and time-points (2, 4, 6, 8 weeks), Dunnett’s procedure (based on the baseline) can avoid false-positive findings. Moreover, as our primary outcome is evaluated by comparing differences between baseline and Week 6, our study results will also be determined by our primary outcome results.

4. The sample size calculation is based on an effect size of 0.75, but it is unclear whether this estimate is conservatively justified. If the true effect size is smaller, the study may be underpowered to detect meaningful differences. Providing a sensitivity analysis or justifying the robustness of this assumption would help address this concern.

-> Thank you for your careful consideration which can affect to detect the differences in our study. Our effect size was calculated by results on our preliminary clinical trials of OJS plus SMS in GERC, which were found to be 0.75. As our sample size was calculated by this exact effect size, it is unlikely that the number of subjects was underestimated. We also added reference of our preliminary clinical trial.

Review2:

1- NO LINE NUMBERS/ NO conclusion, so please add a paragraph at the end of discussion to summarize what are the outcomes from this study for clarity.

-> Thank you for your opinion to add line numbers and conclusion. As we thought line numbers will automatically generated, we had not put line numbers in our manuscript. We had added line numbers and conclusion at the end of discussion. Thank you for your opinion for our paper structure once again.

2- The study makes use of traditional herbal formulations (Ojeok-san and Saengmaek-san), and repeatability may be impacted by the variations in herbal composition.

-> Thank you for that you had made concerns on repeatability which can be raised for herbal medicines. To avoid this issue, we had clearly listed the compositions of Ojeok-san and Saengmaek-san at Table 1. These herbal formulations are manufactured by Han Kook Shin Yak Pharm Co. Ltd., which obtained authorization from Korea Good Manufacturing and quality data of our investigational medicine (OJS and SMS) had also been approved by Ministry of Food and Drug Safety of Korea consisting. Additionally, both of OJS and SMS are insurance-covered herbal medicines packaged in powder package being frequently used in Korea. Insurance-covered medicines are managed by the Ministry of Food and Drug Safety of Korea for their quality every year. Regarding these regulations in Korea, we think the repeatability of herbal medicines will not be the issue for our trial. We once again appreciate your opinion.

3- Despite seeking to evaluate safety and efficacy, the study doesn't seem to look at the mechanisms underlying GERC or the methods that OJS+SMS work.

-> Thank you for your considerations of suggesting mechanism research in this study. We had also considered conducting mechanism research of OJS plus SMS on GERC. However, as our intervention is complex of two herbal medicines, which are already used for GERD and cough, and not a newly developed medicine, our aim was not focused on mechanisms of this complex herbal medicines. As each of OJS and SMS has already revealed their pharmacological mechanisms, we had addressed each of the potential mechanism of OJS and SMS in introduction and discussion part, stating that OJS to regulate GI motility and SMS to attenuate airway inflammation, thereby OJS plus SMS can improve the symptoms of cough induced by GERD. However, in our next trial, we consider to evaluate mechanisms of OJS plus SMS evaluating related mechanisms of GERC.

4- Given that GERC can be chronic and variable, a 4- to 6-week course of treatment and an 8-week total follow-up may not be enough to evaluate long-term efficacy, safety, and recurrence of symptoms.

-> Thank you for your opinion on our trial period, which can be considered short for chronic cough. When designing our trial, we had reviewed related trials and guidelines to detect efficacy of our intervention on GERC. According to the “ERS guidelines on the diagnosis and treatment of chronic cough in adults and children”, the guideline panel considered that it was preferable to undertake sequential therapeutic trials of each agent for 1 to 4 weeks depending on their pharmacology, meaning that medications is considered effective when they had shown significant responses after administration of 1 to 4 weeks in chronic cough [1]. Moreover, in our preliminary trial, OJS plus SMS were found to have significant efficacy on week 4 and 6, but not at week 8, by showing the difference between OJS plus SMS and placebo decreased. At week 8(follow-up period), the total CDS score had been increased in OJS plus SMS group again, showing that the treatment effect was no longer sustained [2]. Thus, we designed with the same intervention period, which can show the overall effectiveness in OJS plus SMS in GERC. We once again appreciate your opinion suggesting us longer follow-up period, and we will consider for our further study.

Reference)

1. Morice, Alyn H., et al. "ERS guidelines on the diagnosis and treatment of chronic cough in adults and children." European Respiratory Journal 55.1 (2020).

2. Lyu, Yee Ran, et al. "Efficacy and safety of ojeok-san plus saengmaek-san for gastroesophageal reflux-induced chronic cough: a pilot, randomized, double-blind, placebo-controlled trial." Frontiers in Pharmacology 13 (2022): 787860.

5- Because the study was only carried out in Korea, its applicability to Western populations or other healthcare environments may be limited by cultural, dietary, and medical variations.

-> Thank you for your consideration to being applied for worldwide population. As your opinion, our study may have limitations on application in other countries. Despite this limitation, we had followed the global regulation for clinical trials. The study protocol will be conducted in accordance with the Declaration of Helsinki and Good Clinical Practice guidelines and complied with the Standard Protocol Items: Recommendations for Interventional Trials (SPIRIT) and Consolidated Standards of Reporting Trials guidelines (CONSORT). Reflecting your opinion, after we conduct this trial in Korea for first, then we will try to expand to perform in other Western countries. Thank you for your thoughtful consideration.

6- There is no reference to the use of objective diagnostic tools for GERD/GERC, such as pH monitoring and impedance testing, which could impair the accuracy of determining the severity of the illness and its response to treatment.

-> Thank you for your recommendation of using multichannel intraluminal impedance pH monitoring, which is known as objective and more accurate diagnostic tool for GERC. As you had mentioned, MII-ph testing are being used to diagnose GERD and its role in chronic cough, thereby can also being used to investigate pharmacological mechanisms. However, we determined MII-ph testing is not essential in our study, as our study does not aim to find a relationship between acid or non-acid reflux and the occurrence of cough, but rather to find differences between symptomatic and non-symptomatic patients with GERC. Moreover, American College of Chest Physicians guidelines (2023) still suggest diagnostic algorithm by excluding other potential causes and we followed this guideline which is also being frequently used in real world clinics. By following guidelines, we had included GERC patients for those who had diagnosed for GERD by endoscopy. We also evaluate GERDQ, which has been reported as an alternative for MII-ph. Although MII-ph testing is more accurate test for diagnosing GERC, following American College of Chest Physicians guidelines or evaluating with GERDQ, can also be used for diagnosis for GERC both on real-world practices or clinical trials without addressing misclassification bias. We had also added the references and explanations for this methodological choice in discussion. Despite these limitations of using MII-ph testing in this trial, we will consider using it on our further trials when evaluating pharmacological mechanisms of OJS plus SMS on GERC. We really appreciate your opinion to strengthen our methodology.

---

## [Decision Letter · Decision Letter 1]

Efficacy, safety, and economic evaluation of Ojeok-san plus Saengmaek-san for gastroesophageal reflux-induced chronic cough: protocol for a randomized, double-blind, placebo-controlled, parallel, multicenter, investigator-initiated clinical trial

PONE-D-24-59964R1

Dear Dr. Lyu,

We’re pleased to inform you that your manuscript has been judged scientifically suitable for publication and will be formally accepted for publication once it meets all outstanding technical requirements.

Kind regards,

Emmanuel O Adewuyi, BPharm, MPH, PhD

Academic Editor

PLOS ONE

Additional Editor Comments (optional):

Reviewers' comments:

Reviewer's Responses to Questions

**Comments to the Author**

1. Does the manuscript provide a valid rationale for the proposed study, with clearly identified and justified research questions?

Reviewer #1: Yes

Reviewer #2: Yes

2. Is the protocol technically sound and planned in a manner that will lead to a meaningful outcome and allow testing the stated hypotheses?

Reviewer #1: Yes

Reviewer #2: Yes

3. Is the methodology feasible and described in sufficient detail to allow the work to be replicable?

Reviewer #1: Yes

Reviewer #2: Yes

4. Have the authors described where all data underlying the findings will be made available when the study is complete?

Reviewer #1: Yes

Reviewer #2: Yes

5. Is the manuscript presented in an intelligible fashion and written in standard English?

Reviewer #1: Yes

Reviewer #2: Yes

You may also provide optional suggestions and comments to authors that they might find helpful in planning their study.

Reviewer #1: The authors effectively revised the paper in response to the feedback received, addressing my concerns with clarity and incorporating suggested improvements to enhance the overall quality of the manuscript.

Reviewer #2: Dear authors,

I have carefully reviewed the revised version and am pleased to see that the authors have addressed all of my previous comments satisfactorily.

I appreciate your response!

**Do you want your identity to be public for this peer review?** For information about this choice, including consent withdrawal, please see our Privacy Policy

Reviewer #1: No

Reviewer #2: No

---

## [Editor Report · Acceptance letter]

PONE-D-24-59964R1

PLOS ONE

Dear Dr. Lyu,

I'm pleased to inform you that your manuscript has been deemed suitable for publication in PLOS ONE. Congratulations! Your manuscript is now being handed over to our production team.

Kind regards,

on behalf of

Dr. Emmanuel O Adewuyi

Academic Editor

PLOS ONE